# Features of Masticatory Behaviors in Older Adults with Oral Hypofunction: A Cross-Sectional Study

**DOI:** 10.3390/jcm11195902

**Published:** 2022-10-06

**Authors:** Chikako Hatayama, Kazuhiro Hori, Hiromi Izuno, Masayo Fukuda, Misao Sawada, Takako Ujihashi, Shogo Yoshimura, Shoko Hori, Hitomi Togawa, Fumiko Uehara, Takahiro Ono

**Affiliations:** 1Division of Comprehensive Prosthodontics, Faculty of Dentistry & Graduate School of Medical and Dental Sciences, Niigata University, Niigata 951-8514, Japan; 2Department of Oral Health Sciences, Faculty of Nursing and Health Care, BAIKA Women’s University, Osaka 567-8578, Japan; 3Department of Oral Health Science, Faculty of Health Science, Kobe Tokiwa University, Kobe 653-0838, Japan

**Keywords:** masticatory behaviors, number of chews, oral hypofunction, masticatory performance, oral function

## Abstract

Although many studies have shown the relationships between oral function and nutrition and health, few reports have investigated the masticatory behaviors of older people. This study aimed to clarify the relationships between oral function and the masticatory behaviors and features of masticatory behaviors with oral hypofunction. A total of 98 community-dwelling independent older adults participated. Seven oral conditions related to oral hypofunction were examined, and the masticatory behaviors when consuming a rice ball were measured. The participants were divided into two groups according to the criteria for oral hypofunction, and the masticatory behaviors were compared. Furthermore, the relationship between masticatory performance and the number of chews was investigated. The chewing rate of the oral hypofunction group was slower than that of the no oral hypofunction group, but there was no difference in the number of chews and chewing time. The decreased tongue–lip motor function group showed a slower chewing rate, and the decreased tongue pressure group showed a smaller number of chews and shorter chewing time. No significant correlation was observed between masticatory performance and behavior. In conclusion, older adults with oral hypofunction chewed slowly due to decreased dexterity, while, even if oral and masticatory function decreased, no compensatory increase in the number of chews was observed.

## 1. Introduction

Various studies to date have shown relationships between healthy life expectancy and oral function [1,2], and between oral function and low nutrition, sarcopenia [3,4], and frailty [5]. In response to this, the Japanese Society of Gerodontology proposed the disease concept of “oral hypofunction” [6], with the aim of preventing severe disease through early diagnosis and the appropriate management of decreased oral function. Seven sub-symptoms and diagnostic criteria were indicated for oral hypofunction: poor oral hygiene, oral dryness, reduced occlusal force, decreased tongue–lip motor function, decreased tongue pressure, decreased masticatory function, and deterioration of swallowing function.

Mastication is the process of grinding food and mixing it with saliva to produce a bolus that can be readily swallowed, and it takes place through the coordination of the various components of oral function [7]. Masticatory function has mainly been evaluated from the perspective of how strongly people can bite (occlusal force) and how efficiently they can crush or mix food (masticatory performance) [8]. At the same time, it was recently pointed out that masticatory behaviors, such as how many times food is chewed and how much time is spent chewing, is important in the prevention of lifestyle-related diseases, such as obesity [9], and aspiration/choking [10]. Such diversification of viewpoints in the evaluation of mastication is essential when addressing the issue of masticatory function in elderly people.

For example, even when a person’s masticatory performance or oral function has declined, it would surely be possible to reduce the risk of poor nutrition or choking if there were a greater number of chews at regular mealtimes so that a diverse range of foods could be ingested. In addition, it may perhaps be the case that elderly people with reduced masticatory performance and oral function compensate by increasing the number of chews. However, there have been almost no reports of studies investigating how masticatory behaviors, such as the number of chews, relates to oral function and masticatory function in elderly persons, and the reality of masticatory behaviors in elderly persons with reduced oral function is unknown.

The purpose of the present study was to observe the masticatory behaviors of community-dwelling elderly people when they ingested a prescribed amount of food, in order to explore the relationships of masticatory behaviors with decreased oral function and masticatory performance. We hypothesized that masticatory performance is negatively correlated with the number of chews, and that elderly people with decreased oral function have a greater number of chews, and we attempted to validate this hypothesis.

## 2. Materials and Methods

### 2.1. Participants

The present study was designed as an exploratory cross-sectional survey. The participants were 98 elderly people (33 men, 65 women; mean age 74.8 ± 6.3 years) living independently in the community in M City. Recruitment was conducted in senior health classes, and the survey was conducted at the venue of the health classes between September 2018 and October 2021. The inclusion criteria were age 60 years or over, living independently, and a participant in the senior health classes sponsored by the municipal government of M City. The exclusion criteria were history of cerebrovascular disorder, dementia, neuromuscular disease, or head and neck tumor, medication related to oral hypofunction, toothache, significant movement teeth due to severe periodontitis, and missing data. We calculated sample size based on the correlation between number of chews on rice ball and mastication performance. Eight-two participants were required for 80% power, with effect size of 0.5, with a two-sided alfa level of 0.05, for correlation (G*Power 3.1.9.7, Heinrich-Heine-Universität, Düsseldorf, Germany). The objectives and methods of the study were fully explained to the participants, who provided their written, informed consent. The present study was carried out with the approval of the Ethics Committee of Niigata University (approval no. 2017-0230).

### 2.2. Survey Items

At first, a dentist performed an intraoral examination to investigate the number of remaining teeth, occlusal support, and the presence or absence of dentures. The number of people living together was asked as a social factor. The survey items were the seven items related to oral hypofunction [6], including (1) poor oral hygiene, (2) oral dryness, (3) reduced occlusal force, (4) decreased tongue-lip motor function, (5) decreased tongue pressure, (6) decreased masticatory function, and (7) deterioration of swallowing function, to which was added an eighth item, (8) measurement of masticatory behaviors. For all items, denture wearers were evaluated with their dentures in place.

#### 2.2.1. Poor Oral Hygiene

This was evaluated using a bacterial counter (PHC Corporation, Osaka, Japan) [11]. A sterile cotton swab was dipped in distilled water and then rubbed on the tongue at a distance of 1 cm from the center of the dorsum of the tongue using a low-pressure specimen collection device, with the rubbing pressure set at 20 gf. The cotton swab was placed in a measuring cup, and the total number of microorganisms was counted by the bacterial counter. A total microbial count of 6.5 Log10 (CFU/mL) or higher (level 4 or higher) was considered to indicate poor oral hygiene.

#### 2.2.2. Oral Dryness

This was evaluated using an oral moisture checking device (Mucus, Life Co., Ltd., Koshigaya, Japan) [12]. The level of mucosal wetness was measured at the center of the dorsum of the tongue, approximately 10 mm from the apex. For measurement, a dedicated sensor cover was fitted to the sensor, and the sensor was held against the test surface for around 2 s with pressure of about 200 g applied to ensure uniform contact. Measurements were taken three times, and a median value of less than 27.0 was considered to indicate oral dryness.

#### 2.2.3. Occlusal Force

Occlusal force was analyzed using a pressure-sensitive sheet (Dental Prescale II, GC Corporation, Tokyo, Japan) [13] and an analysis device (Bite Force Analyzer, GC Corporation, Tokyo, Japan). The pressure-sensitive sheet was placed in the mouth, and the participant was instructed to clench the teeth for 3 s in the maximum intercuspal position. An occlusal force of less than 500 N was considered to indicate reduced occlusal force.

#### 2.2.4. Tongue–Lip Motor Function

The speed and dexterity of tongue and lip movements were comprehensively evaluated using oral diadochokinesis. Participants were required to pronounce each of the syllables /pa/, /ta/, and /ka/ for 5 s, and the number of pronunciations per second of each syllable was measured using an automatic measuring device (Kenko-kun Handy, Takei Scientific Instruments Co., Ltd., Niigata, Japan) [14]. A count of fewer than 6 repetitions of any of the syllables /pa/, /ta/, or /ka/ per second was considered to represent decreased tongue–lip motor function.

#### 2.2.5. Maximum Tongue Pressure

Maximum tongue pressure was measured using a digital probe for tongue pressure measurement (JMS tongue pressure measuring device TPM-01, JMS Co., Ltd., Hiroshima, Japan) [15]. A balloon fitted to the tongue pressure probe was placed against the anterior part of the palate, and the participant was instructed to voluntarily squash the balloon against the palate using the tongue with maximum force for 7 s. After the participant first practiced and then rested to avoid fatigue, measurements were carried out three times, and the mean value was calculated. Maximum tongue pressure of less than 30 kPa was considered to indicate decreased tongue pressure.

#### 2.2.6. Masticatory Performance

Masticatory performance was measured using a test gummy jelly (UHA Mikakuto Co., Ltd., Osaka, Japan) [16]. The participant was instructed to chew a test gummy jelly (5.50 ± 0.05 g) 30 times and then spit it out onto a gauze, and the condition of the comminuted gummy jelly was compared to a 10-stage visual scale and scored from 0 to 9. A score of 0–2 was considered to indicate decreased masticatory performance.

In addition, the fragments of comminuted gummy jelly were placed in a prescribed box (inner dimensions 140 mm × 95 mm × 36 mm) with black markers (7 mm × 7 mm, distance between a markers-width of 88 mm, length 133 mm), and the increase in surface area of the comminuted gummy jelly was calculated using an imaging method [17].

#### 2.2.7. Swallowing Function

Swallowing function was measured using a swallowing screening questionnaire, the 10-item eating assessment tool (EAT-10) [18]. The participant was required to fill in the questionnaire, and a score of 3 or more was considered to indicate deterioration of the swallowing function.

#### 2.2.8. Masticatory Behaviors

Masticatory behaviors were measured using a device for counting the number of chews (bitescan^®^, Sharp Corporation, Sakai, Japan; Figure 1) with dedicated software [19]. This device is designed to be worn on the right auricle with an ear hook, which is available in three sizes (S, M, L). The size that best fits the auricle of the participant was selected to enable the built-in sensor to sense the measurement site behind the auricle. For measurement, a Bluetooth connection with a smartphone (SHM05, Sharp Corporation, Sakai, Japan) was confirmed, the bitescan^®^ with the selected ear-hook of appropriate size was placed on the right ear, and calibration was carried out. For the assessment of masticatory behaviors, the participant was asked to eat a 100-g rice ball (seaweed-rolled rice ball, Marusan Co., Ltd., Higashi-Osaka, Japan); no special eating instructions were given. The participants were simply asked to eat a single rice ball as they normally would, and the measurement was carried out until the rice ball was completely swallowed. Measurements were taken at least 2 h after meals.

As parameters of masticatory behaviors, the number of chews, the number of chews per bite, the chewing rate, and the total chewing time were evaluated. The masticatory behavior items were defined as follows:Number of chews (no.): The total number of chewing cycles during the time to eat 1 rice ball.Number of chews per bite (no.): Mean number of chews per bite which is an uptake action.Chewing rate (no./min): The number of chews per minutes calculated by dividing by total chewing time.Total chewing time (s): The time taken to eat 1 rice ball.

### 2.3. Analysis

Each participant was examined in accordance with the criteria for the seven sub-symptoms of oral hypofunction, and they were considered to have oral hypofunction if they had three of the seven items [6]. The participants were divided into two groups based on the oral hypofunction diagnostic criteria and the criteria for each of the seven sub-symptoms of oral hypofunction, and the masticatory behaviors of each group were compared using the Mann–Whitney U test. The relationship between masticatory performance and the number of chews was examined using Spearman’s correlation coefficient. SPSS Statistics 23.0 (IBM) was used for statistical analysis, and the significance level was set at *p* < 0.05.

## 3. Results

### 3.1. Participants’ Oral Condition and Oral Hypofunction

A total of 113 individuals applied for this study, and 101 met the inclusion and exclusion criteria. Three participants were excluded from the analysis due to withdrawal of consent, and finally 98 participants were included in the analysis.

The number of remain teeth, occlusal status, and usage of removable denture of the participants are shown in Table 1. In addition, the age, height, body weight number of people living together are also presented in Table 1.

Of the 98 participants, 71 (23 men, 48 women, 75.8 ± 6.2 years) had oral hypofunction, and 27 (10 men, 17 women, 72.5 ± 6.0 years) did not (Table 1). Those with oral hypofunction were significantly older than those without, with oral hypofunction present in 32 (65.3%) early-stage elderly people (aged 65–74, 49 persons) and 39 (80.0%) late-stage elderly people (aged 75 and older, 49 persons). The group of participants with poor oral hygiene (91 persons, 92.9%) was largest, and those with deterioration of swallowing function (10 persons, 10.2%) were smallest in number (Table 2).

### 3.2. Comparison of Masticatory Behaviors in Participants with and without Oral Hypofunction

The number of chews (mean ± standard deviation {median}) was 236 ± 103 {233} for participants overall, with no significant difference between the oral hypofunction group (230 ± 89 {240}) and the non-oral hypofunction group (254 ± 133 {228}) (*p* = 0.975) (Table 2).

The number of chews per bite was 32.7 ± 22.1 {26.6} for participants overall, with no significant difference between the oral hypofunction group (30.7 ± 20.3 {26.5}) and the non-oral hypofunction group (37.9 ± 26.0 {29.3}) (*p* = 0.259).

The chewing rate was 77.5 ± 15.0 {78.0} cycles/min for participants overall, with the oral hypofunction group (75.4 ± 13.9 {77.0} cycles/min) significantly slower than the non-oral hypofunction group (83.1 ± 16.4 {81.0} cycles/min) (*p* = 0.035).

The total chewing time was 177 ± 69 {169} s for participants overall, with no significant difference between the oral hypofunction group (176 ± 62 {173} s) and the non-oral hypofunction group (180 ± 87 {156} s) (*p* = 0.477).

### 3.3. Comparison of Masticatory Behaviors by Oral Hypofunction Sub-Symptoms

The analysis of masticatory behaviors by each sub-symptom of oral hypofunction is shown in Table 2.

In the decreased tongue pressure group (*n* = 43), the number of chews (255 ± 97 {262}) was significantly greater than in the non-decreased tongue pressure group (*n* = 55) (221 ± 106 {225}), and total chewing time (194 ± 68 {193} s) was significantly longer than in the non-decreased tongue pressure group (164 ± 68 {155} s).

In the decreased tongue–lip motor function group (*n* = 55), the chewing rate (73.7 ± 14.2 {75.0} cycles/min) was significantly slower than in the non-decreased tongue–lip motor function group (*n* = 43) (82.4 ± 14.6 {82.5} chews/min).

For all other sub-symptoms, there were no significant differences in masticatory behaviors between the groups with and without the sub-symptom.

### 3.4. Relationship between Masticatory Performance and Number of Chews

No significant correlation was found between masticatory performance (amount of increase in surface area of comminuted gummy jelly) and the number of chews when consuming a rice ball (Figure 2; r = 0.055, *p* = 0.600).

No significant correlation was found between masticatory performance (amount of increase in surface area of comminuted gummy jelly) and the number of chews when consuming a rice ball (Spearman’s correlation coefficient: *r* = 0.055, *p* = 0.600).

## 4. Discussion

The present study is the first attempt to objectively measure masticatory behaviors and masticatory performance in independent, community-dwelling, older adults to explore the relationship between the two and also to examine how they are affected by oral hypofunction. The results showed that older adults with oral hypofunction, and in particular elderly people with decreased tongue–lip motor function, have a slower chewing rate, and that older adults with decreased tongue pressure have a greater number of chews and longer total chewing time. In addition, no significant correlation was seen between masticatory performance and the number of chews. These findings show the relationship between oral function and masticatory behaviors in older adults, providing basic data for approaches to dealing with those with declining oral function.

### 4.1. Measurement of Masticatory Behaviors

To extend healthy life expectancy, it is essential to prevent elderly people from falling into the cycle of frailty through low nutrition [20]. It has been reported that people with an inadequate ability to form a food bolus tend to avoid fibrous foods or meat, thus losing variety from their diet [21]. Formation of the food bolus within the mouth requires not only conservation of the remaining teeth, but also maintenance of oral functions, such as tongue function and occlusal force. Various methods of assessment of oral function have, therefore, been devised to enable the management of oral function in older adults.

The prevention of frailty in older adults needs to be approached not just from the perspective of oral function, but also through the assessment of regular dietary behavior and provision of appropriate guidance. For example, it has been reported in edentulous individuals that prosthodontic treatment alone does not result in sufficient improvement in diet and nutritional intake, and the treatment needs to be accompanied by guidance on food selection and nutritional intake [22]. At the same time, masticatory behaviors, such as number and rate of chews at mealtimes, are important for the selection of a wide range of foods and safe swallowing through appropriate bolus formation. While there have been various educational campaigns encouraging people to eat slowly and chew their food thoroughly [23], little has been known to date about the extent to which older adults masticate their food when ingesting.

One reason for the scarcity of reports on the number of chews is the difficulty of accurate and convenient measurement. Studies to date have used a device to measure jaw movement [24] or a muscle activity meter [25,26] to measure the number of chews, but such devices are cumbersome and require measurement to be carried out in a laboratory, so they are unsuitable for surveying large numbers of people. Studies have also been carried out by measuring the number of chews through direct observation of participants at mealtimes or by video recording mealtimes and then measuring masticatory behaviors [27]. However, such measurement environments may differ from regular mealtimes.

We studied the masticatory behaviors using bitescan^®^, a wearable device that measures the number of chews [19]. This device is simply placed on the right ear, and it monitors the changes in the skin surface behind the auricle that accompany masticatory movements. It is connected to a smartphone via Bluetooth to measure parameters relating to masticatory behaviors, so there is no need to restrain the participant in any way. We previously confirmed that the device has sufficient measurement accuracy [19], and we used it to study masticatory behaviors in healthy adults [28,29].

In the present study, masticatory behaviors were measured using rice balls. Rice balls are among the most popular and frequently consumed foods in Japan, and as well as being made at home, they are readily available from supermarkets and convenience stores. It has been reported that the ingestion method, such as the eating utensils used, affects the number of chews [30], but rice balls are generally eaten out of the hands. Furthermore, it is difficult to evaluate the number of chews per bite with a small amount of food that can be ingested in one bite. In the present study, rice balls were used because there is little influence from participants’ preferences or the method of ingestion, rice balls need to be masticated, there is a reasonable amount to be ingested, and measurements can be made under uniform conditions. Based on measurements of 99 healthy adults, it was reported that the number of chews, the number of bites, the number of chews per bite, and the chewing rate with a single rice ball used in the present study all show significant positive correlations with the same parameters during usual meals for an entire day [29].

### 4.2. Oral Hypofunction

Of the 98 participants in the present study, 72.4% had oral hypofunction. Looking at prior studies of oral hypofunction [4,5,31,32,33,34], the reported prevalence shows a wide range, from 43% [31] to 63% [32,34]. The participants of these studies included not just different age groups, but also community-dwelling elderly persons [4,5,31,33,34] and older adults who were visiting a dental clinic [32], with the incidence in community-dwelling older adults often reported in the range of 50–60%. The results of the present study seem to show a somewhat high proportion of participants with oral hypofunction. However, all prior reports indicate that the incidence of oral hypofunction increases with age [31,32,34], and the same trend was seen in the present study. In addition, looking at the sub-symptoms, it may be seen that the incidences of poor oral hygiene [33,34] and decreased tongue–lip motor function [4,31,32,34] are often high. In the present study as well, there were many older adults with poor oral hygiene, which may have resulted in a higher incidence of oral hypofunction. In the case of participants who were visiting a dental clinic, it could be assumed that they had some kind of oral complaint [32]. However, in the case of older adults who visited a dental clinic for a regular check-up, they may be less likely to have oral hypofunction as a result of dental treatment or poor oral hygiene thanks to their oral hygiene management. In the present study, the participants were independent older adults aged 60 years or older who participated in the senior health classes sponsored by the municipal government of M City, and not patients who were visiting a dental clinic. However, though there were a few participants who complained of problems with dentures or poor oral status, there were also participants receiving regular oral hygiene care at their regular dental clinic. The present study did not investigate dental visit history or oral symptoms, but there is a need for studies that survey these items in order to adjust the participants for analysis in the future.

### 4.3. The Characteristics of Masticatory Behaviors in Older Adults with Oral Hypofunction (and Its Sub-Symptoms)

The results of the present study showed the number of chews (median value) to be 240 in the oral hypofunction group and 228 in the non-oral hypofunction group, with no significant difference between the two. This result suggests that there is no increase in the number of chews to compensate for the decline in oral function. At the same time, the number of chews per bite in the oral hypofunction group was 26.5, which was slightly less than the number in the non-oral hypofunction group (29.3), although the difference was not significant. In addition, though the chewing rate was significantly slower in the oral hypofunction group than in the non-oral hypofunction group, there was no significant difference between the oral hypofunction group and the non-oral hypofunction group in the total chewing time to eat 1 rice ball. It therefore appears that, even though older adults with oral hypofunction masticated a rice ball more slowly, they had fewer chews per bite, with the result that there was no difference between the oral hypofunction group and the non-oral hypofunction group in the overall number of chews or total chewing time. This suggests that bolus formation may be inadequate in older adults with oral hypofunction, which would indicate a risk of choking or incomplete absorption of nutrients. It appears that older adults with oral hypofunction not only need dental treatment and improvement or management of oral function, but also guidance in taking their time over meals and chewing their food well.

Older adults with decreased tongue–lip motor function showed a significantly slower chewing rate than those without this sub-symptom. Decreased tongue–lip motor function is a condition in which the speed and dexterity of tongue and lip movements are reduced due to neuromuscular system dysfunction. This results in incomplete bolus formation and spills during mastication, which impact negatively on mastication and swallowing, consequently limiting the types and amount of food that can be ingested. In addition, decreased tongue–lip motor function can cause problems with articulation and speech. Speech disorders in older adults can lead to social deterioration, not just because they cause difficulties in communication, but also because they can lead to the affected person being reluctant to meet others or avoiding going out. Since participants in the decreased tongue–lip motor function group had a slower chewing rate, total chewing time would be expected to be longer. However, there was no difference in total chewing time between the decreased tongue–lip motor function group and the non-decreased group. It may therefore be conjectured that, even though these participants were chewing slowly, they either swallowed the food soon without adequate mastication, or else there was a large amount of food per bite.

In addition, in the decreased tongue pressure group, the number of chews (262) and the total chewing time (193 s) were significantly greater than in the non-decreased tongue pressure group (number of chews: 225, total chewing time: 155 s). The tongue plays an important role in mastication, swallowing, and pronunciation, and the measured values for these items decrease when the muscle strength of the suprahyoid muscle group declines [35]. In particular, dexterity of tongue movement and muscle strength are essential at each stage of the masticatory and swallowing process, starting with transport of food taken into the anterior part of the mouth to the molars (stage I transport), followed by the mixing of food fragments crushed by the molars with saliva, formation of the bolus (processing), transport of the bolus to the oropharynx (stage II transport), and then the subsequent ejection of the bolus and maintenance of swallowing pressure when the swallowing reflex occurs [36]. There have been prior reports of the relationships of decreased tongue pressure to activities of daily living (ADL) [37] and dysphagia [38]. It has also been reported that decreased tongue pressure is associated with longer mealtimes [39] and the intake of formula diet [40], suggesting that decreased tongue pressure also affects the form of food that can be ingested. In the present study as well, elderly people with decreased tongue pressure showed an increased number of chews and longer total chewing time in order to form a bolus adequately, suggesting that decreased tongue pressure affects bolus formation.

### 4.4. Relationship between Masticatory Behaviors and Masticatory Performance

Since masticatory performance was thought to be closely related to masticatory behaviors, in the present study, masticatory performance was evaluated not only by the score method, but also by an imaging method to evaluate the increase in surface area of comminuted gummy jelly in order to give a detailed evaluation. With the evaluation of masticatory performance using a test gummy jelly, the significant relationship between the score method and imaging was previously demonstrated [17,41]. In the present study, no significant correlation was found between increased surface area of the comminuted gummy jelly and number of chews of a rice ball.

To date, there have only been a few studies investigating the relationship between masticatory performance and number of chews. Some of these have reported no association [42,43,44], which is the same result as the present study, but one study reported a negative association, with a greater number of chews in participants with low masticatory performance [45].

Many of these reports had participants with a wide age range, from young to old [44,45]. In addition, a variety of foods was used for evaluation, such as peanuts [42,43], carrots [45], and gummy jelly [44], and it therefore appears that the results are greatly influenced by the food.

Although the measurement conditions in the survey may have influenced the results, attention should be given to the finding that there are elderly people with both low masticatory performance and low number of chews. Such elders are at risk of choking, as well as defective digestion or absorption of nutrients, suggesting the need to not only restore occlusal support and improve the oral environment through dental treatment, but also to pay attention to their daily masticatory behaviors and, where necessary, provide guidance on chewing habits.

### 4.5. Limitations and Future Study

In this study, we did not investigate the degree of periodontal disease or the history of dental caries though the applicant who had toothache or significant movement teeth were excluded. Even if there is no pain, caries and periodontal disease might affect chewing behavior. Although it is thought that the participants’ oral condition lacks homogeneity, in this study we analyzed oral function as an explanatory variable.

In addition, the bitescan^®^ should have been used for a full day or several days to assess daily masticatory behaviors. However, since some older adults have difficulty using a smartphone or wearable device, masticatory behaviors were evaluated with a single rice ball at the research site. If improvements could be made to the bitescan^®^ to allow simple self-monitoring by older adults, we would like to conduct surveys of masticatory behaviors at everyday meals.

In the future, a large survey of mastication behaviors would further clarify the details of mastication behaviors according to oral conditions and age in the elderly persons. We also want to investigate the relationship between mastication behaviors and nutritional status. Furthermore, we plan to examine the effects of mastication behaviors modification on the health of the elderly people.

Although these are possible limitations, the present study is the first to investigate masticatory behaviors in older adults in detail through the number of chews, chewing time, and chewing rate. In addition, it was possible to clarify characteristics, such as the slower chewing rate in older adults with decreased oral function. We believe that these results show some of the detailed aspects of mastication in older adults with decreased oral function. Elderly people with decreased oral and masticatory functions should be given masticatory instruction in addition to rehabilitation and dental treatment. The results of this study are considered to be useful as indices for mastication instruction.

## 5. Conclusions

In older adults living independently in the community, the chewing rate of the oral hypofunction group was significantly slower than that of the non-oral hypofunction group, but no difference was observed between the groups in the number of chews or total chewing time. In particular, the decreased tongue–lip motor function group showed a significantly slower chewing rate, and the decreased tongue pressure group showed a significantly higher number of chews and significantly longer total chewing time. These results indicate that decline in oral function affects masticatory behaviors. At the same time, older adults with decreased oral or masticatory function showed no compensatory increase in the number of chews, suggesting that functional decline may increase the risk of choking or affect digestion and the absorption of nutrients. These results suggest the need for guidance on mastication that covers individual oral functions.

## Figures and Tables

**Figure 1 jcm-11-05902-f001:**
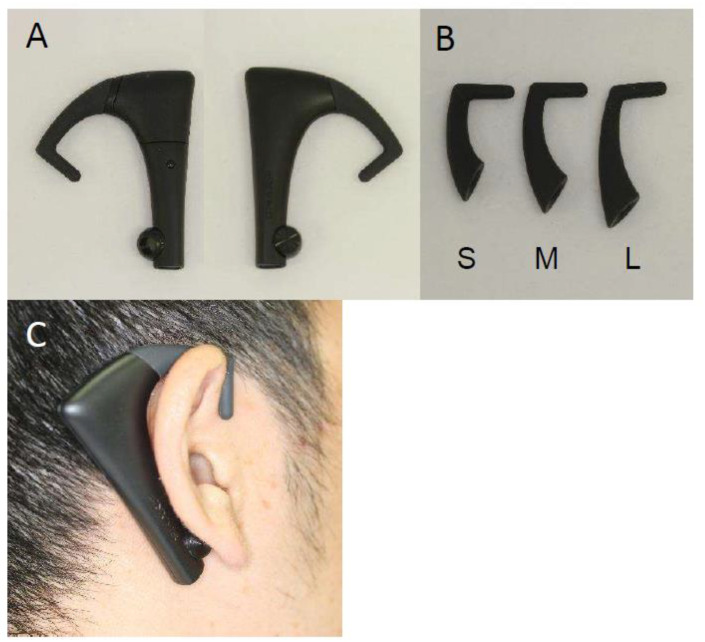
A wearable device for counting the number of chews (bitescan^®^). (**A**) Main unit; (**B**) ear-hooks for size adjustment; (**C**) bitescan^®^ when worn.

**Figure 2 jcm-11-05902-f002:**
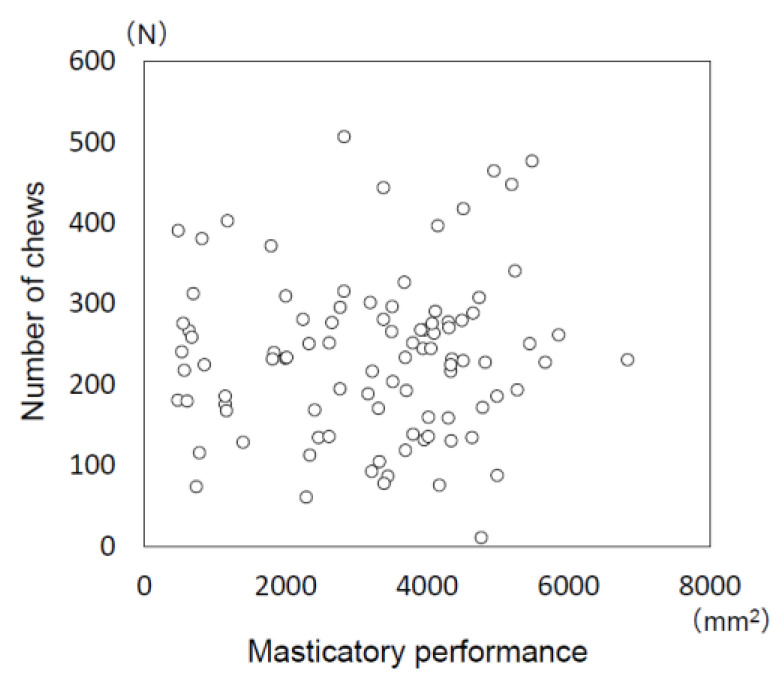
Relationship between masticatory performance and the number of chews when consuming a rice ball.

**Table 1 jcm-11-05902-t001:** Information of the participants.

			All		Male		Female	
		*n* (%)	98	(100)	33	(33.7)	65	(66.3)
Age (y)		mean (SD)	74.8	(6.3)	74.6	(6.1)	75.0	(6.5)
Oral hypofunction		*n* (%)	71	(72.4)	23	(69.7)	48	(73.8)
Height (cm)		mean (SD)	156.7	(3.3)	165.1	(2.0)	148.4	(4.7)
Body weight (kg)		mean (SD)	59.4	(1.5)	63.8	(3.3)	55.0	(1.8)
Number of remain teeth		mean (SD)	21.6	(7.8)	22.1	(6.9)	21.4	(8.3)
Occlusal status	Eichner A	*n* (%)	53	(54.2)	15	(45.5)	38	(58.5)
	Eichner B	*n* (%)	22	(22.9)	9	(27.2)	13	(20.0)
	Eichner C	*n* (%)	23	(22.9)	9	(27.2)	14	(21.5)
N of participants using removal denture		*n* (%)	38	(38.8)	14	(42.4)	24	(36.9)
N of people living together		*n* (SD)	2.2	(1.2)	2.3	(1.3)	2.1	(1.1)

**Table 2 jcm-11-05902-t002:** The masticatory behaviors in consuming a rice ball between the participants with and without oral hypofunction and subcategory.

		*n*	Number of Chews (Cycles)			Number of Chews Per Bite (Cycles)			Chewing Rate (Cycles/min)			Total Chewing Time (s)		
			Median	IQR	P *	Median	IQR	P *	Median	IQR	P *	Median	IQR	P*
Oral hypofunction	Yes	71	240	(171–280)	0.975	26.5	(17.2–38.8)	0.259	77.0	(66.8–85.0)	0.035	173	(131–216)	0.477
	No	27	228	(160–341)		29.3	(19.0–51.0)		81.0	(75.0–92.0)		156	(121–226)	
Poor oral hygiene	Yes	91	232	(168–280)	0.327	27.0	(18.0–44.0)	0.644	78.7	(70.7–88.0)	0.200	165	(127–207)	0.161
	No	7	262	(169–313)		24.0	(22.0–28.2)		72.1	(66.0–81.0)		210	(156–258)	
Oral dryness	Yes	56	230	(161–274)	0.208	27.2	(18.7–38.5)	0.892	79.5	(73.3–87.8)	0.277	163	(128–195)	0.088
	No	42	251	(171–308)		26.2	(18.8–45.5)		75.9	(65.0–84.5)		186	(125–252)	
Reduced occlusal force	Yes	43	234	(159–281)	0.747	27.0	(17.2–40.8)	0.652	77.0	(65.0–85.9)	0.235	170	(129–216)	0.836
	No	55	232	(171–281)		26.2	(19.0–45.0)		80.0	(72.0–88.0)		169	(126–223)	
Decreased tongue pressure	Yes	43	262	(180–297)	0.046	27.0	(21.0–44.0)	0.506	77.9	(70.7–83.0)	0.266	193	(151–242)	0.010
	No	55	225	(139–268)		23.2	(18.6–40.8)		80.0	(68.0–90.5)		155	(121–192)	
Decreased tongue-lip motor function	Yes	55	230	(139–277)	0.229	17.1	(16.8–43.6)	0.594	75.0	(66.0–83.0)	0.003	169	(127–210)	0.783
	No	43	234	(180–316)		19.0	(18.7–40.8)		82.5	(75.0–90.7)		164	(129–223)	
Decreased masticatory function	Yes	15	241	(180–313)	0.653	25.7	(19.6–32.4)	0.531	74.3	(66.0–84.0)	0.354	169	(131–250)	0.421
	No	83	232	(160–281)		27.0	(18.6–44.7)		79.0	(70.7–88.0)		169	(125–202)	
Deterioration of swallowing function	Yes	10	251	(218–267)	0.651	29.7	(22.3–47.4)	0.439	80.5	(63.7–91.8)	0.972	186	(154–212)	0.372
	No	88	232	(162–287)		26.4	(18.2–39.4)		78.0	(71.0–86.2)		164	(126–221)	

*: Mann–Whitney’s U test, IQR: interquartile range.

## Data Availability

The data presented in this study are available on request from the corresponding author. The data are not publicly available due to ethical restrictions.

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
