# Peer review of "Features of Masticatory Behaviors in Older Adults with Oral Hypofunction: A Cross-Sectional Study"

_jcm, 2022, doi:10.3390/jcm11195902_

Round 1

Reviewer 1 Report

In this study, Hatayama et al. conducted an exploratory cross-sectional study to explore the relationships of masticatory behaviors with decreased oral function and masticatory performance. While this study presents some positive characteristics, my view is that the study does not present innovative results. The hypothesized results are too obvious considering the vast literature available. Furthermore, the methodology presents major flaws as they did not account for missing teeth, existing periodontal diseases (that may interfere with proprioception) or even active dental caries/history of caries (e.g., ICDAS or DMFT). Missing these oral conditions resulted in suboptimally  adjusted results.

Other major concerns:

- There is some confusion on Material and Methods and Results. You should describe the flowchart of patients in the results.

- The inclusion criteria is not comprehensive. Why focusing just on patients 60 years old or older? What about medication that could cause (as side effects) some of the so-called "decreased oral function and masticatory performance"?

- I advise you to use the STROBE guideline for reporting.

I am sorry to not bring better news. Yet, I hope these could strengthen your study for a future submission.

Author Response

Reviewer 1

In this study, Hatayama et al. conducted an exploratory cross-sectional study to explore the relationships of masticatory behaviors with decreased oral function and masticatory performance. While this study presents some positive characteristics, my view is that the study does not present innovative results. The hypothesized results are too obvious considering the vast literature available. Furthermore, the methodology presents major flaws as they did not account for missing teeth, existing periodontal diseases (that may interfere with proprioception) or even active dental caries/history of caries (e.g., ICDAS or DMFT). Missing these oral conditions resulted in suboptimally adjusted results.

Ans.

First of all, we appreciate you for your kind review and valuable comments. As mentioned in the introduction, no devices have been established to accurately and simply measure human chewing habits and chewing behavior. Therefore, it is true that there is very little scientific evidence regarding the relationship between oral function and chewing behavior, which you consider to be “too obvious”. Therefore, we published a paper on the accuracy verification of bitescan in 2021, and later on chewing behavior and health conditions (Yoshimura S, et al., Sci Rep, 2022, Uehara F, et al. , JMIR Mhealth Uhealth, 2022). These works certainly verified what had long been considered self-evident, but we believe that they were innovative in that they provided objective and quantitative measurement data that had never existed before. Also, the fact that our papers have been accepted is proof of that.

As you pointed out, we also think that the number of teeth and the presence or absence of dentures are very interesting background factors for chewing behavior in the elderly. The same goes for the extent of periodontal disease and caries, which we have not investigated. However, the theme of this paper is to investigate the relationship between oral health status and mastication behavior using diagnostic items based on oral hypofunction in older people. Based on the results of this study, we would like to create an index of chewing behavior that is appropriate for individual older people by investigating background factors in a larger population. In consideration of your point, we clearly state that this research is positioned as a limitation.

Other major concerns:

- There is some confusion on Material and Methods and Results. You should describe the flowchart of patients in the results.

Ans.

We described the detail of participants in the Results as follows.

113 applied for this study, and 101 met the inclusion and exclusion criteria. Three participants were excluded from the analysis due to withdrawal of consent, and finally 98 participants were included in the analysis.

- The inclusion criteria is not comprehensive. Why focusing just on patients 60 years old or older? What about medication that could cause (as side effects) some of the so-called "decreased oral function and masticatory performance"?

Ans.

The Japanese Society of Geriatric Dentistry has proposed the concept and diagnostic criteria for oral hypofunction, citing the possibility of malnutrition in elderly people with decreased oral function. Several reports now suggest that even younger adults are at risk for oral hypofunction. In this study, we investigated the characteristics of masticatory behaviors in elderly people aged 60 years and older who are at risk of oral hypofunction.

In addition, in this study, we excluded the elderly with a history of diseases that could clearly impair oral function and those with medication related to oral hypofunction. We added a note about this to the method.

- I advise you to use the STROBE guideline for reporting.

Ans. 

Thank you for your useful advice. We have checked the STROBE guidelines and attached a check sheet.

I am sorry to not bring better news. Yet, I hope these could strengthen your study for a future submission

Ans.

Thank you for your careful review. We revised the manuscript according to your advice. We hope that you will understand and accept the significance of this manuscript.

Reviewer 2 Report

The authors evaluated the symptoms of oral hypofunction and correlated them to the masticatory behavior in relation to the number of chewing strokes and  duration of chewing the test food. The research is interesting and may have significance in prosthodontic treatment of elderely subjects. 

Although authors  evaluated and described in detail oral hypofunction symptoms and masticatory perfomance analyse, I must raise a huge concern about the homogenity of the study group.  There were no information about the dental status of the participants. Were they complete denture or removable partial denture  wearers. Or did they have fixed partial denture or natural dentition, with how many occlusal functional units. The lacking data in inclusion criterion have a huge impact on interepretation of the results, concerning  occlusal bite intensity and masticatory perfomance. 

Author Response

Reviewer 2

The authors evaluated the symptoms of oral hypofunction and correlated them to the masticatory behavior in relation to the number of chewing strokes and duration of chewing the test food. The research is interesting and may have significance in prosthodontic treatment of elderly subjects. 

Ans.

We appreciate you for your interest in our research and for recognizing its importance. Also, thank you for your kind review. We have read your comments carefully and made corrections.

Although authors evaluated and described in detail oral hypofunction symptoms and masticatory perfomance analyse, I must raise a huge concern about the homogenity of the study group.  There were no information about the dental status of the participants. Were they complete denture or removable partial denture wearers. Or did they have fixed partial denture or natural dentition, with how many occlusal functional units. The lacking data in inclusion criterion have a huge impact on interepretation of the results, concerning occlusal bite intensity and masticatory perfomance. 

Ans.

Thank you for your important remarks. In this study, participants with toothache or significant movement due to severe periodontitis were excluded. However, we did not consider occlusal support or the number of remaining teeth in the participant selection criteria. We added to the Limitation about the lack of homogeneity of participants in the condition of the teeth and the usage of dentures of the participants. Additionally, we added a new Table 1 with information on the participants' number of remaining teeth and the use of dentures.

Reviewer 3 Report

Dear Authors,

Congratulations for your work! The article is very well written, it provides sufficient information regarding masticatory behaviors in older adults with oral hypofunction.

Even though is a very well known fact that the elderly patients have oral hypofunction, the article clarify the relationships between oral function and masticatory behaviors and features of masticatory behaviors with oral hypofunction.

I have minor recommendations:

line 50: which is the lifestyle-related disease?

line 95: does the measurements take into consideration the variables such as moment of the day, before or after the meals etc?

You used for the measurements rice balls.  If other aliments where used, the results would be different?

It was a pleasure to read this study, and I think that the results can be used for further studies that can be focused in prevention of the elder patients frailty due to oral hypofunction.

Author Response

Reviewer 3

Dear Authors,

Congratulations for your work! The article is very well written, it provides sufficient information regarding masticatory behaviors in older adults with oral hypofunction.

Even though is a very well known fact that the elderly patients have oral hypofunction, the article clarify the relationships between oral function and masticatory behaviors and features of masticatory behaviors with oral hypofunction.

Ans.

We appreciate you for your interest in our research and for recognizing its importance. Also, thank you for your kind review. We have read your comments carefully and made corrections.

line 50: which is the lifestyle-related disease?

Ans.

We revised as “the lifestyle-related disease such as obesity”

line 95: does the measurements take into consideration the variables such as moment of the day, before or after the meals etc?

Ans.

We did not specify the time of measurement, but we measured after 2 hours or more after eating. We added this issue at 2.2.8.

You used for the measurements rice balls.  If other aliments where used, the results would be different?

Ans.

We used rice balls for the measurements. The reason is described in L259.

If other food is used, the number of chewing times may be different, but we do not have the data. However, we have shown in the past report that the number of chews of rice balls had a weak correlation with the number of chews per day, and we believe that the trend obtained this time will not change.

It was a pleasure to read this study, and I think that the results can be used for further studies that can be focused in prevention of the elder patients frailty due to oral hypofunction.

Thank you again for your kind comments. We are very encouraged by your comments.

Reviewer 4 Report

First, thank you for giving me the opportunity to review this manuscript. It is an interesting study but needs changes:

Material and methods

-Describe the setting, locations, and relevant dates, including periods of recruitment, and data collection

- Was it considered if the participants were missing teeth? How many? What type of oral rehabilitation did they have?

- Was it considered if the participants were taking any type of medication that could alter the study variables?

- Were the socioeconomic variables of the patients considered?

- STROBE Statement-Checklist of items that should be included in reports of cross-sectional studies. This list specifies that, although it is a convenience sample, the sample size must be calculated. Therefore, explain how the study size was arrived at.

Results

-By using such a wide age range, it might be interesting to clarify in a table whether there are differences between age ranges with each study variable.

Discussion

- The limitations of the study should be much more developed, as there are many more than those mentioned above.

- There should be a section on the practical application of this study.

- Describe future lines of research

Author Response

Reviewer 4

First, thank you for giving me the opportunity to review this manuscript. It is an interesting study but needs changes:

We appreciate you for your interest in our research. Also, thank you for your kind review. We have read your comments carefully and made corrections.

Material and methods

-Describe the setting, locations, and relevant dates, including periods of recruitment, and data collection

Ans.

Thank you for your important remarks.

We described these points in 2.1. Participants section.

- Was it considered if the participants were missing teeth? How many? What type of oral rehabilitation did they have?

Ans.

We added a new Table 1 with information on the participants' number of remaining teeth and the use of dentures. Participants did not receive any type of oral rehabilitation.

- Was it considered if the participants were taking any type of medication that could alter the study variables?

Ans.

We investigated the participants' medical history and medications in recruitment, and excluded participants with a history of diseases or taking medications that may reduce oral function.

- Were the socioeconomic variables of the patients considered?

Ans.

We investigated participants' socioeconomic factors only by the number of family members living together. The number of family members living together is described in the new Table 1. Other socioeconomic factors were not considered.

- STROBE Statement-Checklist of items that should be included in reports of cross-sectional studies. This list specifies that, although it is a convenience sample, the sample size must be calculated. Therefore, explain how the study size was arrived at.

Ans.

We calculated sample size based on the correlation between number of chews on rice ball and mastication performance. Eight-two participants were required for 80% power, with effect size of 0.5, with a two-sided alfa level of 0.05, for correlation (G*Power 3.1.9.7).

Results

-By using such a wide age range, it might be interesting to clarify in a table whether there are differences between age ranges with each study variable.

Ans.

Thank you for your insightful comments. It is certainly an interesting analysis to examine differences in study variables by age group. However, since this research did not focus on the difference of age ranges and the number of samples is small, we would like to examine the characteristics of each age group in the future.

Discussion

- The limitations of the study should be much more developed, as there are many more than those mentioned above.

Ans.

We have responded to all reviewers' comments and added some issues to the limitations.

- There should be a section on the practical application of this study.

Ans.

We added sentences on practical application and future plans to the last paragraph of the discussion.

- Describe future lines of research

Ans.

We added sentences on practical application and future plans to the last paragraph of the discussion.

Round 2

Reviewer 2 Report

I have no other comments to add.